# GlycoTAIL and FlexiTAIL as Half-Life Extension Modules for Recombinant Antibody Fragments

**DOI:** 10.3390/molecules27103272

**Published:** 2022-05-19

**Authors:** Oliver Seifert, Roland E. Kontermann

**Affiliations:** 1Institute of Cell Biology and Immunology, University of Stuttgart, 70569 Stuttgart, Germany; roland.kontermann@izi.uni-stuttgart.de; 2Stuttgart Research Center Systems Biology (SRCSB), University of Stuttgart, 70569 Stuttgart, Germany

**Keywords:** scFv, diabody, scFv-EHD2 fusion protein, CEA, FAP, hydrodynamic radius, half-life extension, PEG mimetic

## Abstract

Many therapeutic proteins are small in size and are rapidly cleared from circulation. Consequently, half-life extension strategies have emerged to improve pharmacokinetic properties, including fusion or binding to long-lasting serum proteins, chemical modifications with hydrophilic polymers such as PEGylation, or, more recently, fusion to PEG mimetic polypeptides. In the present study, two different PEG mimetic approaches, the GlycoTAIL and the FlexiTAIL, were applied to increase the hydrodynamic radius of antibody fragments of different sizes and valencies, including scFv, diabody, and scFv-EHD2 fusion proteins. The GlycoTAIL and FlexiTAIL sequences of varying lengths are composed of aliphatic and hydrophilic residues, with the GlycoTAIL furthermore comprising N-glycosylation sites. All modified proteins could be produced in a mammalian expression system without reducing stability and antigen binding, and all modified proteins exhibited a prolonged half-life and increased drug disposition in mice. The strongest effects were observed for proteins comprising a FlexiTAIL of 248 residues. Thus, the GlycoTAIL and FlexiTAIL sequences represent a flexible and modular system to improve the pharmacokinetic properties of proteins.

## 1. Introduction

Antibodies have found numerous applications for diagnosis, prophylaxis, and therapy of diseases [1]. While most of the approved therapeutic antibodies belong to the IgG class, recent developments have also focused on the use of antibody fragments [2,3]. Thus, Fc-less antibody fragments such as monospecific Fab or scFv, bispecific tandem-scFv, and antibody fusion proteins have been developed and approved for various applications, including the treatment of hematologic, vascular, and neurological disorders, and the engagement of T-cells in cancer therapy [4,5,6]. Furthermore, conjugates of antibody fragments with payloads or imaging reagents have been developed for therapeutic and diagnostic applications [7,8,9]. Although their small size facilitates tissue penetration, these molecules encounter rapid elimination, mainly by renal filtration. Consequently, for therapeutic applications, antibody fragments are used for short-term treatment, employed locoregionally, or administered as continuous infusion when used systemically [10].

To facilitate and improve therapeutic use, several strategies have been developed to increase the half-life of small antibody fragments, including chemical conjugation of polyethylene glycol (PEG) chains (PEGylation) and fusion to moieties capable of binding to long-circulating serum proteins such as albumin and immunoglobulins [11,12]. More recently, various approaches were established to use artificial polypeptide chains as PEG mimetics, i.e., with PEG-like half-life extending properties, circumventing the need for chemical coupling and providing biocompatible and biodegradable polymers [13]. These approaches include PEG mimetics such as SAPA [14], HRM [15], GLK [16], ELP [17], XTEN [18] and PAS [19]. Similar to PEG, these PEG mimetics allow for adjusting the pharmacokinetic properties to the therapeutic need by adapting the length of the fused polypeptide chain [20].

In a previous study, we developed a polypeptide chain comprising several N-glycosylation sites (GlycoTAIL) as a half-life extension module [21]. This module was applied to extend the half-life of a bispecific single-chain diabody (scDb) possessing a molecular mass of approximately 50 kDa and rapid renal clearance. The scDb-GlycoTAIL protein exhibited, compared, e.g., to PEGylated scDb and an scDb fused to an albumin-binding domain, a moderately improved half-life, drug exposure, and tumor accumulation [22].

Here, we have now adapted this approach by using non-glycosylated derivatives of the GlycoTAIL sequence of varying lengths (FlexiTAILs). The sequence of these FlexiTAILs consists of randomly arranged aliphatic and hydrophilic residues. These FlexiTAILs were fused to various recombinant antibody derivatives of different sizes and valencies, including monovalent scFvs and bivalent diabody (Db) and scFv-EHD2 fusion proteins [23]. All molecules were produced in a mammalian expression system and analyzed for purity, antigen-binding activity, and pharmacokinetic properties after i.v. injection into mice. Furthermore, we included cysteine-modified derivatives of the scFv-EHD2 fusion proteins for a defined coupling of fluorophores. These experiments demonstrated that all modified antibody fragments can be produced and that the pharmacokinetic parameters can be adjusted using either GlycoTAIL or FlexiTAILs of varying lengths.

## 2. Results

### 2.1. Generation of Antibody Fragments Comprising GlycoTAILs and FlexiTAILs

To generate half-life extended antibody derivatives, a previously described GlycoTAIL (GT7) [21] comprising seven N-glycosylation sites in a stretch of fifty-five amino acids, or a novel non-glycosylated derivative thereof (FlexiTAIL) were fused to the C-terminus of the antibody chains. For the GlycoTAIL module, the N-glycosylation sites are located six to sixteen amino acids from each other. Both modules are based on a random sequence of glycine, alanine, threonine, serine, asparagine, glutamine, and aspartate. The FlexiTAIL building block of 62 amino acid residues was either used two times (FlexiTAIL124; FT124) or four times (FlexiTAIL248; FT248) (Figure 1). Antibody fragments are directed against carcinoembryonic antigen (CEA) and included a single-chain Fv fragment (scFv), a bivalent diabody (Db), and a bivalent scFv-EHD2 [23] fusion protein with calculated molecular masses of 26.9 kDa, 56.7 kDa, and 83.0 kDa, respectively (Figure 1, Table 1, Table 2 and Table 3). In all molecules, the half-life extension modules were fused to the C-terminus of the polypeptide chains. Thus, the scFv comprised one module and the Db and scFv-EHD2 two modules. The antibody proteins were produced in mammalian HEK293-6E suspension cells and purified with the N-terminally located His-Tag using immobilized metal affinity chromatography (IMAC).

### 2.2. Half-Life Extension of an scFv

ScFvCEA-GT7 and two different scFv-FT fusion proteins (scFvCEA-FT124 and scFvCEA-FT248) were produced with yields of 0.7 mg/L for the scFvCEA-FT124 and 1.7 mg/L for the scFvCEA-FT248, while lower yields were obtained for the unmodified scFvCEA (0.12 mg/L) and scFvCEA-GT7 (0.16 mg/L). SDS-PAGE analysis of the molecules showed under reducing and non-reducing conditions similar bands with one single band for at ~30 kDa for scFvCEA, ~55 kDa for scFvCEA-FT124, and ~70 kDa scFvCEA-FT248, while the scFvCEA-GT7 showed a panel of bands between 40 and 50 kDa, indicating heterogenous N-glycosylation of scFvCEA-GT7. In size-exclusion chromatography (SEC) analysis, the scFvCEA showed the expected size of ~27.1 kDa (R_S_ = 2.3 nm), while the other molecules showed an increased hydrodynamic radius corresponding to apparent molecular masses of 53.6 to 87.0 kDa (R_S_ = 3.0–3.4 nm) for scFvCEA-GT7, 91.9 kDa (R_S_ = 3.6 nm) for scFvCEA-FT124, and 178.9 kDa (R_S_ = 4.6 nm) for scFvCEA-FT248 (Figure 2C, Table 1). Dynamic light scattering (DLS) revealed for all molecules a similar aggregation temperature of 46 to 47 °C (Appendix A).

The antigen-binding activity was analyzed by ELISA using CEA as an immobilized antigen (Figure 2D). All molecules showed a concentration-dependent binding with EC_50_ values between 0.5 to 0.8 nM. The scFvCEA-FT248 molecule showed significantly reduced binding to the antigen compared to the other analyzed molecules (scFvCEA (*), scFvCEA-GT7 (*), scFv-FT128 (**)), most likely due to the sterical hindrance of binding the antigen. Binding was further analyzed by flow cytometry using the CEA^+^ colorectal cancer cell line LS147T (Figure 2E). All molecules showed similar binding to the cells with EC_50_ values of 1.1 to 1.7 nM. Thus, modification of the scFv with GlycoTAIL or FlexiTAILs did not affect antigen-binding activity.

Pharmacokinetic properties of the antibody molecules were studied in immunocompetent CD1 mice. Proteins were applied i.v. into the tail of mice and serum samples were analyzed in ELISA, i.e., detecting functional molecules (Figure 2F). As expected, the scFvCEA molecules were rapidly cleared from the blood system with a terminal half-life of ~0.6 h, while the modified scFv molecules demonstrated a statistically significant prolonged terminal half-life. Thus, scFvCEA-GT7 exhibited a terminal half-life of 1.4 h, while the addition of the FlexiTAILs prolonged the terminal half-life to 5.7 (****) and 11.5 (****) hours for scFvCEA-FT124 and scFvCEA-FT248, respectively, which correlated with increased drug exposure (AUC) of the half-life extended molecules, up to 2-, 5-, and 16-fold for scFvCEA-GT7 (***), scFvCEA-FT128 (****), and scFvCEA-FT248 (****), respectively (Table 1). These data confirmed that the increased hydrodynamic radius of the modified scFv observed in SEC translated to increased terminal half-lives.

### 2.3. Half-Life Extension of a Bivalent Diabody

The effects of fusion of the GlycoTAIL and the FlexiTAILs were then studied for a bivalent diabody, i.e., a non-covalently linked dimer of a V_H_-V_L_ chain with a five amino acid long linker connecting the two variable domains, exhibiting twice the size of an scFv (Figure 3A). In SDS-PAGE analysis, a major band of ~31 kDa was observed for DbCEA, while DbCEA-FT124 and DbCEA-FT248 showed a band at ~51 kDa and ~70 kDa, respectively. Again, for the DbCEA-GT7 several bands in the range between 41 to 56 kDa were observed under reducing and non-reducing conditions (Figure 3B). The increase in molecular masses was confirmed by SEC. Here, the modified Db molecules exhibited an R_S_ of 3.1 nm corresponding to a molecular mass of 56.8 kDa, while for DbCEA-GT7 an R_S_ of 3.9 nm (98.7 kDa) and DbCEA-FT124 and DbCEA-FT248 R_S_ of 3.8 nm (93.3 kDa) and 4.9 nm (156.5 kDa), respectively, were determined (Figure 3C). Thus, the addition of the GlycoTAIL or the FlexiTAILs led to a strongly increased hydrodynamic radius. The aggregation temperatures of the modified molecules were not affected or even slightly increased (47 °C for DbCEA and 48 to 50 °C for the modified diabodies) (Appendix A).

In ELISA experiments, the unmodified DbCEA bound in a concentration-dependent manner to immobilized CEA with an EC_50_ value of ~0.6 nM, while the modified molecules bound with EC_50_ values in the range of 1.3 to 1.7 nM (Figure 3D). Significant differences were detected for the DbCEA compared to all other analyzed molecules (DbCEA-GT7 (***), DbCEA-FT128 (***), and DbCEA-FT248 (****)). In flow cytometry studies using LS174T cells, all Db-based molecules showed similar binding to the cells with EC_50_ values in the range of 1.3 to 1.5 nM (Figure 3E). The configuration with GlycoTAIL and FlexiTAIL did not interfere with binding to CEA in ELISA and flow cytometry analysis. 

The analysis of the pharmacokinetic properties showed a clearly statistically prolonged half-life for the DbCEA-FT248 compared to the DbCEA, DbCEA-GT7, and DbFT128 molecules (****). Thus, the terminal half-life of DbCEA-FT248 was 5.7 h compared to 1.7 h for the unmodified DbCEA, which translated into a five-fold increased AUC of DbCEA-FT248 compared to DbCEA (Figure 3F) (Table 2).

### 2.4. Half-Life Extension of a Bivalent scFv-EHD2 Fusion Protein

Next, the half-life extension modules were applied to a bivalent scFv-EHD2 molecule. Here, the EHD2 (IgE heavy chain domain 2) moiety serves as a covalently-linked dimerization module [23,24] (Figure 4A). The EHD2 is derived from the Cε2 domain of IgE, comprises a single N-glycosylation site, and forms homodimers. Again, molecules were generated in HEK293-6E mammalian cells and purified via Ni-NTA chromatography from the supernatant. SDS-PAGE analysis showed under reducing conditions two bands for scFvCEA-EHD2 at 42–45 kDa indicating heterogeneity of N-glycosylation of the EHD2 domain. The scFvCEA-EHD2-GT7 showed, as seen for scFvCEA-GT7 and DbCEA-GT7, a ladder of bands between 65 to 75 kDa. Single bands at ~70 kDa or ~100 kDa were detected for scFvCEA-EHD2-FT124 or scFvCEA-EHD2-FT248 molecule, respectively, under reducing conditions (Figure 4B). Under non-reduced conditions, a strong band at ~100 kDa was observed for scFv-EHD2, corresponding to the covalently linked dimer. Some further bands were observed at 38 to 40 kDa, corresponding to non-covalently linked scFvCEA-EHD2 chains. The apparent molecular mass of the modified scFvCEA-EHD2 molecules was strongly increased, with ~200 kDa for scFvCEA-EHD2-GT7 and for scFv-EHD2-FT124, and >200 kDa for scFvCEA-EHD2-FT248. This was confirmed by SEC, which revealed for the scFvCEA-EHD2 molecule an R_S_ of 4.2 nm (~110 kDa), an R_S_ of 6.4 nm (244.4 kDa), 6.4 nm (~255.4 kDa), and 7.7 nm (420.5 kDa) for scFvCEA-EHD2-GT7, scFvCEA-EHD2-FT124, and scFvCEA-EHD2-FT248, respectively (Figure 4C). The thermal stability was not changed compared to the scFvCEA-EHD2 molecule with an aggregation temperature of between 47 to 48 °C (Appendix A). All molecules showed a concentration-dependent binding to CEA in ELISA with similar EC_50_ values between 0.2 and 0.3 nM (Figure 4D). Statistical significance was detected for the scFvCEA-EHD2 molecule compared to the other analyzed molecules (scFvCEA-EHD2-GT7 (***), scFvCEA-EHD2-FT148 (****), and scFvCEA-EHD2-FT248 (**)).

PK studies demonstrated that the increased hydrodynamic radius of the modified scFvCEA-EHD2 molecules also resulted in a prolonged half-life of the modified molecules, which was again most pronounced for the FT248-modified molecule. Here, scFvCEA-EHD2-FT248 exhibited a terminal half-life of 24.9 h, compared to 5.0 h for the unmodified molecule (Figure 4E), corresponding to an approximately five-fold increased drug exposure (AUC). Significant differences were observed in the terminal half-life between the scFvCEA-EHD2 molecule and the other modified molecules (scFvCEA-EHD2-GT7 (*), scFvCEA-EHD2-FT128 (**), and scFvCEA-EHD2-FT248 (****)), while the scFvCEA-EHD2-FT248 was also significant to the scFvCEA-EHD2-GT7 (**) and the scFvCEA-EHD2-FT128 (*). For the AUC analysis, only the scFvCEA-EHD2-FT248 showed significant differences from the other analyzed molecules (****).

### 2.5. Generation of Cysteine-Modified Antibody Molecules

Finally, the FlexiTAIL strategy was applied to a fibroblast activation protein (FAP)-targeting scFv-EHD2 fusion protein using scFvhu36 as an antigen-binding module [25]. An additional cysteine residue was inserted into the linker of the scFv fragment for further chemical labeling of the molecule with a fluorophore (Cy5 Maleimidie Mono Dye). This scFvFAP-Cys was furthermore used to generate a bivalent scFv-EHD2-Cys molecule as well as two FlexiTAIL fusion proteins (scFv-EHD2-FT124-Cys, scFv-EHD2-FT248-Cys) (Figure 5A). All four antibody molecules were produced in HEK293-6E cells and purified by IMAC. Purified proteins were then conjugated with the Cy5 fluorophore (Figure 5A). SDS-PAGE analysis of the molecules showed under reducing conditions one band at ~25 kDa for scFvFAP-Cy5 and ~50 kDa for scFvFAP-EHD2-Cy5 corresponding to the calculated molecular masses (Figure 5B). For the FlexiTAIL-modified molecules, we observed a major band at ~65 kDa for the scFvFAP-EHD2-FT124-Cy5 and ~100 kDa for the scFvFAP-EHD2-FT248-Cy5 molecule. Under non-reducing conditions, the scFv exhibited a similar size as under non-reducing conditions, while covalently linked dimers were observed for the EHD2-based molecules with a major band at 100 kDa for scFv-EHD2-Cy5, ~200 kDa for scFvFAP-EHD2-FT124-Cy5 and above 200 kDa for the scFvFAP-EHD2-FT248-Cy5 molecule. Some additional weaker bands were observed at around 45 kDa, 65 kDa, and 100 kDa for the EHD2 derivatives, indicating non-covalently linked polypeptide chains, caused presumably by the mild reduction used during the conjugation step. For all four molecules, major peaks were observed in SEC, with R_S_ of 2.6 nm (~32 kDa) for scFvFAP-Cy5, 4.4 nm (~111 kDa) for scFvFAP-EHD2-Cy5, 6.3 nm (~265 kDa) for scFvFAP-EHD2-FT124-Cy5 and 7.5 nm (~265 kDa) for scFvFAP-EHD2-FT248-Cy5, respectively. Thus, a strong increase in the hydrodynamic radius was observed for the FlexiTAIL-modified molecules. Some additional minor peaks were observed, presumably due to the conjugation. 

The binding of the labeled molecules to HT1080-FAP cells was analyzed by flow cytometry showing a concentration-dependent binding with EC_50_ values of ~3.3 nM for the monovalent scFvFAP-Cy5 and 0.4 nM for the bivalent scFvFAP-EHD2-Cy5 molecule, while scFvFAP-EHD2-FT124-Cy5 and scFvFAP-EHD2-FT248-Cy5 showed binding with EC_50_ values of 1.1 nM. The difference of EC_50_ of the monovalent (scFv) in comparison to the bivalent (scFv-EHD2, scFv-EHD2-FT128, scFv-EHD2-FT248) binding is significant (****) and most likely based on the avidity effect of the bivalent molecules. As the modified scFvFAP-EHD2-FT-Cy5 molecules also showed significantly worse binding (****) than the unmodified scFvFAP-EHD2-Cy5 molecule, we believe this might be a sterical hindrance of antigen-binding (Table 4).

The pharmacokinetic analysis demonstrated a prolonged terminal half-life of the scFvFAP-EHD2-FT124-Cy5 and the scFvFAP-EHD2-FT248-Cy5 compared to scFvFAP-EHD2, with terminal half-lives of ~36.6 and ~39.3 h compared to 2.6 h for scFvFAP-EHD2-Cy5 and 1.2 h for scFvFAP-Cy5. Significant differences were observed for both scFvFAP-EHD2-FT-Cy5 molecules compared to the unmodified molecules (****). This increased half-life correlated with an increased drug exposure which was a 5.6-fold increase for scFvFAP-EHD2-FT248-Cy5 and a 4.4-fold increase for scFvFAP-EHD2-FT124-Cy5. Here, the scFvFAP-EHD2-FT248-Cy5 showed a significant difference compared to the other analyzed molecules (****) (Table 4). 

## 3. Discussion

In this study, half-life-extended antibody fragments were generated by fusing flexible polypeptide chains of varying lengths and compositions. This included the fusion of a short N-glycosylated polypeptide chain of 52 aa (GlycoTAIL), as well as two post-translationally unmodified flexible polypeptide chains of 124 and 248 residues (FlexiTAILs). All antibody fragments could be produced in mammalian HEK293-6E cells and purified using affinity chromatography. Importantly, the modifications did not or only marginally affect the binding properties and the thermal stability of the molecules as compared to the unmodified molecules. Thus, a shielding effect described for example for some PEGylated proteins and proteins modified with other PEG mimetics such as XTEN was not observed for our PEG mimetics [26,27,28,29]. However, we cannot exclude that longer FlexiTAILs and application to other proteins might affect the bioactivity of the therapeutic moiety. Nevertheless, the two applied FlexiTAIL sequences (FT124 and FT248) are similar in lengths to XTEN sequences used to prolong the half-life of glucagon and hGH, both showing strongly reduced in vitro potency, while antibody fragments fused with FT124 or FT248 showed no reduced binding activity [28,29]. 

Modifications of the antibody fragments with GlycoTAIL and FlexiTAILs resulted in an increased hydrodynamic radius of the molecules compared to the unmodified proteins as demonstrated by size-exclusion chromatography. Of note, for the modified proteins the apparent molecular mass was between a 1.3 to 3.7-fold increase compared to the calculated molecular mass, indicative of the PEG-like properties of the GlycoTAIL and FleixTAILs. This increased hydrodynamic radius translated into prolonged terminal half-lives and increased drug exposure (AUC) in vivo after a single i.v. injection. These effects were strongest for FT248-modified proteins. Thus, the terminal half-life and AUC of the scFv fragment with a molecular mass of 27 kDa was increased approximately 17-fold for the scFv-FT248 derivative, with a calculated molecular mass of 48 kDa but an apparent molecular mass of 179 kDa. For the larger antibody molecules, i.e., the diabody with a molecular mass of 56.7 kDa and the scFv-EHD2 fusion protein with a molecular mass of 83 kDa, effects were less pronounced. Here, the FT248 modification resulted in an approximately 3.3 to 5-fold increased terminal half-life and AUC (Figure 6). This indicates that proteins close to or above the renal filtration threshold benefit to a lesser extent from half-life extending modifications. 

Half-lives of proteins modified with a FlexiTAIL might be further increased by using even longer sequences. Thus, in other studies with recombinant PEG mimetics, such as PAS, sequences with a length of up to 1000 residues were fused to the N- or C-terminus of various biologics [30]. A correlation of the lengths of the fused PEG mimetic with terminal half-life was described [19]. Our FlexiTAILs behaved similarly to these PAS modifications. Thus, a PASylated Fab fragment comprising a PAS chain of 200 residues prolonged half-life from 1.34 h to 5.2 h, which is similar to the half-life extension of the diabody, exhibiting a similar size to a Fab fragment (1.7 h for the unmodified diabody and 5.7 h for the Db-FT248).

In our study, we focused on C-terminal fusions of the GlycoTAIL and FlexiTAILs. Obviously, as for any other recombinant PEG mimetic, the GlycoTAIL and FlexiTAILs can also be fused to the N-terminus or to both ends, or even used as a linker between two protein moieties, thus, allowing great flexibility in the design of half-life extended proteins, including antibody fragments, but also other proteins such as antibody-mimetic scaffold proteins [31,32,33], peptides, hormones, and growth factors [11].

Therapeutic activity is, however, not only influenced by a long half-life but also affected by other properties such as tissue penetration and diffusion. Thus, for an EpCAM-specific DARPin-MMAF conjugate fused to either XTEN or PAS sequences, an intermediate size and half-life of the conjugates showed the strongest anti-tumor effects. The authors concluded that this was a compromise of serum half-life and diffusion within the tumor achieved by fusion of a PAS300 or PAS600 sequence to the small DARPin-drug conjugate [34]. 

The GlycoTAIL and FlexiTAIL chains are composed of seven randomly arranged aliphatic and hydrophilic residues (G, A, N, Q, S, T, D) with approximately 50% being A and G. They differ in their composition from other frequently used PEG mimetic sequences. For example, PASylation is based on random sequences of P, A, and S and XTEN sequences consist of six random amino acids G, P, A, S, T, and E. These two and other PEG mimetics such as GLK, ELP, SAPA and HRM sequences [14,15,16,17] comprise proline, which can adopt a cis or trans conformation but is not required in our GlycoTAIL and FlexiTAIL sequences to increase the hydrodynamic radius and to provide PEG-like properties. 

In addition, the GlycoTAIL sequences comprise N-glycans added during post-translational processes within the cell. Our study demonstrated that expression in mammalian cells results in heterogeneous glycosylation as revealed by SDS-PAGE and SEC analysis. This is in accordance with previous findings using a bispecific single-chain diabody [21]. Here, we also observed heterogeneity in the number of attached N-glycans as well as the composition of the N-glycans. This and the observed reduced productivity might pose some additional challenges for process development. Of note, pharmacokinetic properties of the various antibody fragments were only moderately improved by the addition of the GlycoTAIL, in accordance with findings for the scDb analyzed in the previous study [21,22], and thus might not be the first choice for half-life extension. 

In addition to antibody molecules differing in size and valency, we provide the first evidence that these half-life extension modules can also be applied to generate antibody conjugates, shown here for coupling a fluorescent dye. This approach might be applicable to generate reagents for drug delivery or in vivo imaging studies with adapted half-life to achieve for example favorable tumor to blood ratios [8,35,36,37,38].

In summary, we established the GlyoTAIL and FlexiTAIL moieties as suitable half-life extension modules. The fusion of these moieties increased the hydrodynamic radius of the antibody molecules and resulted in improved pharmacokinetic properties. Thus, GlycoTAIL and especially FelxiTAILs are modular building blocks suitable to adapt the half-life of proteins to the therapeutic needs. 

## 4. Materials and Methods

### 4.1. Materials

For ELISA experiments horse-radish-peroxidas (HRP)-conjugated anti-His antibody was purchased from Santa Cruz Biotechnology (HIS-6 His-Probe-HRP, sc-8036, Heidelberg, Germany). Antibodies for flow cytometry analysis phycoerythrin (PE)-conjugated anti-His antibody was purchased from Miltenyi Biotec (His antibody, PE, 130-120-718, Bergisch Gladbach, Germany). The human colon adenocarcinoma cell line LS174T was purchased from ECACC (Wiltshire, UK) and cultured in RPMI1640 supplemented with 5% FBS. The stably transfected fibrosarcoma HT1080 cell lines expressing human FAP (kindly provided by W. Rettig, Boehringer Ingelheim Pharma, Vienna, Austria) were grown in RPMI, 5% FBS.

### 4.2. Antibody Production and Purification

The DNA sequence of GlycoTAIL and FlexiTAIL were ordered by GeneArt (ThermoFisher Scientific, Karlsruhe, Germany) and cloned together with the antibody fragments into a modified pSecTagA vector. Antibody molecules were transiently transfected into HEK293-6E suspension cells using polyethylenimine (PEI; linear, 25 kDa, Sigma-Aldrich, Darmstadt, Germany) for transfection. HEK293-6E cells were provided by the National Research Council of Canada (Ottawa, Ontario, ON, Canada) and cultivated in F17 Freestyle expression medium (ThermoFisher) supplemented with 0.1% Kolliphor P-118 (Sigma-Aldrich), 4 mM GlutaMAX (ThermoFisher), and 25 µg/mL G418. By adding TN1 (20% tryptone N1 (Organotechnie S.A.S., La Courneuve, France) in F17 medium) to feed the culture 24 h posttransfection, protein production was initiated, and cells were cultivated for additional 4 days at 37 °C. Supernatants were dialyzed against PBS and proteins were purified by immobilized metal affinity chromatography (IMAC) using Ni-NTA (Macherey-Nagel; 745400.100, Düren, Germany). Preparations were dialyzed against PBS at 4 °C. 

### 4.3. Antibody Characterization 

Antibody molecules were analyzed by SDS-PAGE (3 µg) under reducing or non-reducing conditions and stained with Coomassie-Brilliant Blue G-250. Purity and integrity of molecules were analyzed via size-exclusion chromatography (SEC) using a Waters 2695 HPLC in combination with a TSKgel SuperSW mAb HR column (822854, Sigma-Aldrich) at a flow rate of 0.5 mL/min using 0.1 M Na_2_HPO_4_/NaH_2_PO_4_, 0.1 M Na_2_SO_4_, pH 6.7 as mobile phase. Standard proteins: thyroglobulin (669 kDa, R_S_ 8.5 nm), β-amylase (200 kDa, R_S_ 5.4 nm), bovine serum albumin (67 kDa, R_S_ 3.55 nm), carbonic anhydrase (29 kDa, R_S_ 2.35 nm), and Flag-peptide (1 kDa). Stokes radii of antibodies were interpolated from standard proteins.

### 4.4. Enzyme-Linked Immunosorbent Assay (ELISA)

The 96-well plates were coated with the extracellular domain of CEA (ABIN934505) (200 ng/well in PBS) overnight at 4 °C and residual binding site was blocked with 2% (*w*/*v*) skim milk powder in PBS (MPBS, 200 µL/well). The antibodies were diluted in MPBS and titrated 1 to 3 in duplicates starting from 100 nM and incubated for 1 h at RT. Bound antibodies were detected with HRP-conjugated antibodies specific for His-tag (sc-8036; Santa Cruz Biotechnology) in case of the bound antibody molecules. Detection antibodies were incubated for one additional hour at RT. TMB (1 mg/mL; 0.006% (*v*/*v*) in 100 mM Na-acetate buffer, pH 6) was used as substrate, reaction was terminated using 50 µL 1 M H_2_SO_4,_ and absorption was measured at a wavelength of 450 nm. In general, plates were washed three times with PBST (PBS + 0.005% Tween20) and twice with PBS in between each incubation step and in advance of the detection.

### 4.5. Flow Cytometry Analysis

LS174T or HT1080-FAP cells (1 × 10^5^ per well) were incubated with a serial dilution of different antibodies for 1 h at 4 °C. After washing cells twice, bound antibodies were either detected using a PE-labeled anti-His antibody (130-120-787, Miltenyi Biotec, Bergisch Gladbach, Germany) or directly in case of Cy5-labeled proteins. Flow cytometry was performed using MACSQuant Analyzer 10 or MACSQuant VYB (both Miltenyi Biotec). Relative mean fluorescence intensities (MFI) were calculated as followed: rel. MFI = ((MFI_sample_ − (MFI_detection_ − MFI_cells_))/MFI_cells_).

### 4.6. Dynamic Light Scattering

ZetaSizer Nano ZS (Malvern, Worcester, United Kingdom) was used to analyze the thermal stability of the proteins by dynamic light scattering. Purified protein was exposed to increasing temperature (35 °C to 70 °C) in 1 °C intervals with 2-min equilibration steps. The aggregation temperature was defined by the starting point of the increase in the mean count rate.

### 4.7. Chemical Coupling of a Fluorophore

Purified cysteine-containing proteins were reduced by the addition of 8 mM tris(2-carboxyethyl)phosphine (TCEP; 77720; Thermo Fisher) for 10 min at RT. Adequacy amount of Cy5-labeling kit (PA25031; Merck, Darmstadt, Germany) was added to the reduced protein and finally introduced nitrogen to remove any possibility of oxidation of the molecule. After incubation of 2 h at RT, samples were purified with PD-10 columns (17-0851-01; GE Healthcare, Solingen, Germany). 

### 4.8. Pharmacokinetics

Animal care and all performed experiments were in accordance with Federal and European guidelines and have been approved by university and state authorities. A total of 25 µg of proteins were injected into the tail vein of female CD1 mice (Charles River, Freiburg im Breisgau, Germany, three animals per molecule) in a total volume of 100 µL. Blood samples were taken after 3 min, 1 h, 2 h, 6 h, 24 h, 72 h, and 168 h after injection and incubated on ice immediately to obtain serum samples after centrifugation (16,000× *g*, 4 °C, 20 min), which were stored at −20 °C until analysis. Serum concentration of antibodies was determined via ELISA using either CEA (ABIN934505) or FAP-Flag (extracellular domain of FAP: 38-760 aa) as immobilized antigen. Bound antibodies were detected using HRP-conjugated anti-His antibody (sc-8036; Santa Cruz Biotechnology). PK analysis (t_½_α, t_½_β, AUC) was performed using Excel add-ins. The AUC describes the area under the curve and can be used as drug exposure of the applied molecule within the body.

### 4.9. Statistics

All data are represented as mean ± SD. Significances were calculated by GraphPad Prism 7.0.1 (San Diego, CA, USA) and results were compared by one-way ANOVA followed by Tukey’s multiple comparison test (post-test). *p* < 0.05 (*), *p* < 0.01 (**), *p* < 0.001 (***), *p* < 0.0001 (****), n.d. (not determined).

## Figures and Tables

**Figure 1 molecules-27-03272-f001:**
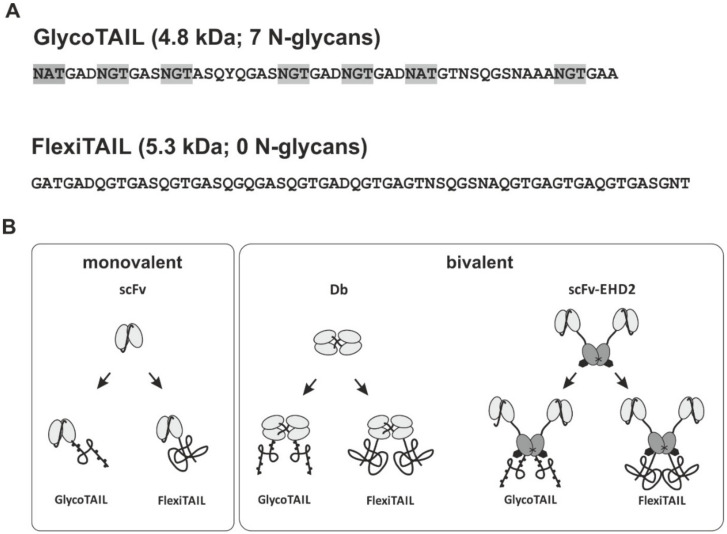
**Sequence and schematic illustration of GlycoTAIL and FlexiTAIL molecules.** (**A**) Sequence of designed GlycoTAIL and FlexiTAIL. The GlycoTAIL moiety consist of 55 amino acids and 7 N-glycan sites, while a single FlexiTAIL moiety consist of 62 amino acids without any N-glycan sites. N-glycan sites are highlighted in gray. (**B**) Schematic illustration of monovalent and bivalent antibody molecules. The scFv and diabody is shown in light gray, EHD2 [23] is shown in gray.

**Figure 2 molecules-27-03272-f002:**
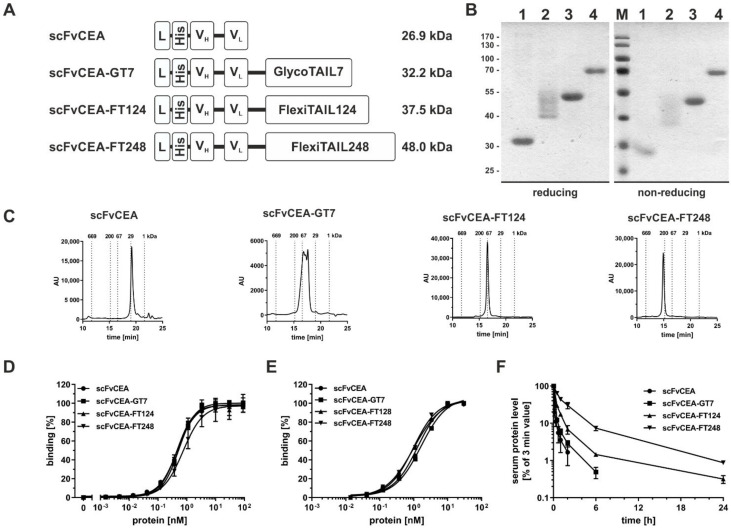
**Analysis of scFvCEA-based antibody molecules.** (**A**) Schematic illustration of the monovalent scFvCEA molecules comprising C-terminal GlycoTAIL or a FlexiTAIL moiety and N-terminal a His-tag. (**B**) SDS-PAGE analysis of the purified scFvCEA molecules (1: scFvCEA, 2: scFvCEA-GT7, 3: scFvCEA-FT124, 4: scFvCEA-FT248) under reducing and non-reducing conditions in a 12% (*m*/*v*) acrylamide gel (3 µg were loaded per lane; M: marker). (**C**) Size-exclusion chromatography of purified scFvCEA molecules. (**D**) Binding of different monovalent scFvCEA molecules to immobilized CEA in ELISA. (**E**) Cell binding of the different scFvCEA molecules to the colon cancer cell line LS147T determined by flow cytometry. Bound antibodies were measured using a PE-conjugated anti-His antibody. (**F**) Pharmacokinetic analysis of different scFvCEA molecules, which were applied intravenously into immunocompetent CD1 mice. Protein amount of blood samples was determined via ELISA experiments. We observed very strong significant differences in terminal half-life between scFvCEA to scFvCEA-FT128 and scFvCEA-FT248 (****) and also strong significant differences between scFvCEA to scFvCEA-GT7 (***). Mean ± SD, *n* = 3.

**Figure 3 molecules-27-03272-f003:**
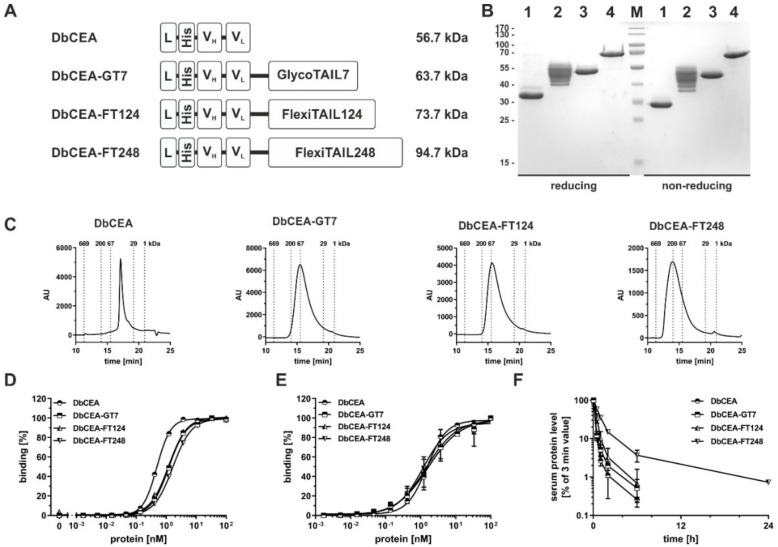
**Analysis of DbCEA-based antibody molecules.** (**A**) Schematic illustration of the bivalent DbCEA molecules comprising C-terminal GlycoTAIL or a FlexiTAIL moiety and/or N-terminal a His-tag. (**B**) SDS-PAGE analysis of the purified DbCEA molecules (1: DbCEA, 2: DbCEA-GT7, 3: DbCEA-FT124, 4: DbCEA-FT248) under reducing and non-reducing conditions in a 12% (*m*/*v*) acrylamide gel (3 µg were loaded per lane; M: marker). (**C**) Size-exclusion chromatography of purified DbCEA molecules. (**D**) Binding of different bivalent DbCEA molecules to immobilized CEA in ELISA. (**E**) Cell binding of the different DbCEA molecules to the colon cancer cell line LS147T determined by flow cytometry. Bound antibodies were measured using a PE-conjugated anti-His antibody. (**F**) Pharmacokinetic analysis of different DbCEA molecules, which were applied intravenously into immunocompetent CD1 mice. Protein amount of blood samples was determined via ELISA experiments. We calculated for the terminal half-life as well as for the drug exposure of the DbCEA-FT248 very strong significant differences compared to all other molecules (****). Mean ± SD, *n* = 3.

**Figure 4 molecules-27-03272-f004:**
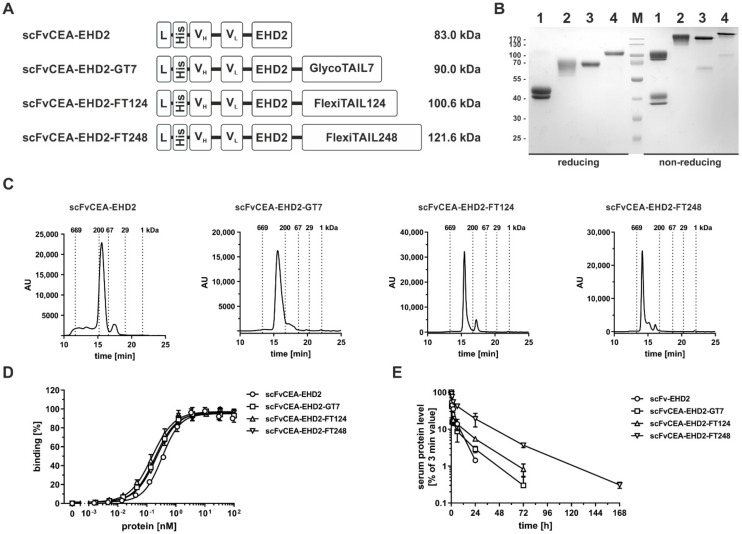
**Analysis of scFvCEA-EHD2-based antibody molecules.** (**A**) Schematic illustration of the bivalent scFvCEA-EHD2 molecules comprising C-terminal GlycoTAIL or a FlexiTAIL moiety and/or N-terminal a His-tag. (**B**) SDS-PAGE analysis of the purified scFvCEA-EHD2 molecules (1: scFvCEA-EHD2, 2: scFvCEA-EHD2-GT7, 3: scFvCEA-EHD2-FT124, 4: scFvCEA-EHD2-FT248) under reducing and non-reducing conditions in a 12% (m/v) acrylamide gel (3 µg were loaded per lane; M: marker). (**C**) Size-exclusion chromatography of purified scFvCEA-EHD2 molecules. (**D**) Binding of different bivalent scFvCEA-EHD2 molecules to immobilized CEA in ELISA. (**E**) Pharmacokinetic analysis of different scFvCEA-EHD2 molecules, which were applied intravenously into immunocompetent CD1 mice. Protein amount of blood samples was determined via ELISA experiments. The unmodified scFvCEA-EHD2 showed significantly reduced terminal half-lives compared to the modified scFvCDEA-GT7 (*), and to both scFvCEA-EHD2-FT molecules (****), while the scFvCEA-EHD2(FT248) also showed significant differences compared to scFvCEA-EHD2-GT7 (**) and scFvCEA-EHD2-FT124 (*). In case of the drug exposure, there was also a significant difference between the scFvCEA-EHD2 molecule compared to all other modified molecules (****). Mean ± SD, *n* = 3.

**Figure 5 molecules-27-03272-f005:**
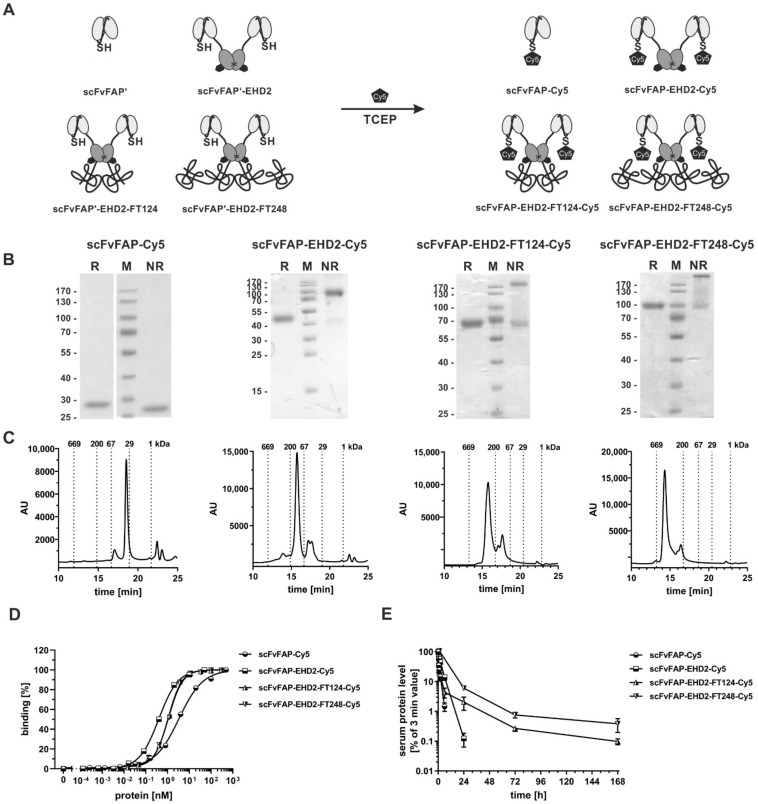
**Analysis of scFvFAP-EHD2-based antibody molecules conjugated with Cy5.** (**A**) Schematic illustration of Cy5-conjugation of the Cys-modified molecules. By the addition of TCEP and melimide-Cy5, fluorophore-labeled molecules were generated. (**B**) SDS-PAGE analysis of the purified anti-FAP molecules (scFvFAP-Cy5, scFvFAP-EHD2-Cy5, scFvFAP-EHD2-FT124-Cy5, scFvFAP-EHD2-FT248-Cy5) under reducing (R) and non-reducing (NR) conditions in a 12% (*m*/*v*) acrylamide gel (3 µg was loaded per lane; M: marker). (**C**) Size-exclusion chromatography of purified scFvFAP-EHD2 molecules. (**D**) Binding of different bivalent scFvFAP-EHD2-Cy5 molecules to HT1080-FAP cells in flow cytometry. (**E**) Pharmacokinetic analysis of different scFvFAP-EHD2-Cy5 molecules, which were applied intravenously into immunocompetent CD1 mice. Protein amount of blood samples was determined via ELISA experiments. In case of the terminal half-life, we calculated strong increased differences between the unmodified molecules scFvFAP and scFvFAP-EHD2 compared to the modified scFvFAP-EHD2-FT molecules (****) and also for the drug exposure for the scFvFAP-EHD2-FT248 compared to all other molecules (****). Mean ± SD, *n* = 3.

**Figure 6 molecules-27-03272-f006:**
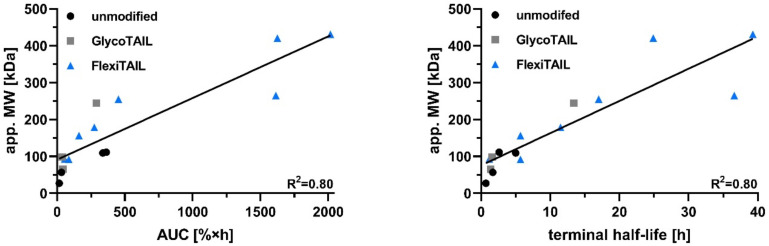
**Analysis of the apparent molecular weight and the PK profile.** The apparent molecular weight was either analyzed against the mean of terminal half-life [h] or against the mean of the drug exposure (AUC [%×h]) of the different molecules. A linear fit was included in the graphs. R^2^ describes the goodness of the fit.

**Table 1 molecules-27-03272-t001:** Biochemical, binding, and pharmacokinetic properties of scFv-based molecules. Mean ± SD, n.d.: not determined, *n* = 3.

	scFvCEA	scFvCEA-GT7	scFvCEA-FT124	scFvCEA-FT248
**calc. MW [kDa]**	26.9	32.2	37.5	48.0
**app. MW [kDa]**	27.1	53.6–87.0	91.9	178.9
**agg. temp. [°C]**	46	n.d.	47	47
**ELISA [nM]**	0.5 ± 0.04	0.5 ± 0.05	0.5 ± 0.05	0.8 ± 0.14
**FACS [nM]**	1.3 ± 0.07	1.7 ± 0.10	1.7 ± 0.95	1.1 ± 0.18
**t_½_α [h]**	0.1 ± 0.01	0.2 ± 0.01	0.4 ± 0.02	0.8 ± 0.09
**t_½_β [h]**	0.7 ± 0.30	1.4 ± 0.09	5.7 ± 0.62	11.5 ± 0.64
**AUC [%×h]**	16 ± 4	42 ± 3	87 ± 4	275 ± 31

**Table 2 molecules-27-03272-t002:** Biochemical, binding, and pharmacokinetic properties of Db-based molecules. Mean ± SD, *n* = 3.

	DbCEA	DbCEA-GT7	DbCEA-FT124	DbCEA-FT248
**calc. MW [kDa]**	56.7	63.7	73.7	94.7
**app. MW [kDa]**	56.8	98.7	93.3	156.5
**agg. temp. [°C]**	47	48	50	49
**ELISA [nM]**	0.6 ± 0.07	1.4 ± 0.19	1.3 ± 0.13	1.7 ± 0.11
**FACS [nM]**	1.5 ± 0.06	1.5 ± 0.26	1.3 ± 0.33	1.5 ± 0.65
**t_½_α [h]**	0.2 ± 0.01	0.2 ± 0.02	0.3 ± 0.06	0.6 ± 0.05
**t_½_β [h]**	1.7 ± 0.30	1.6 ± 0.50	1.2 ± 0.50	5.7 ± 0.50
**AUC [%×h]**	32 ± 2	38 ± 4	56 ± 15	162 ± 16

**Table 3 molecules-27-03272-t003:** Biochemical, binding, and pharmacokinetic properties of scFv-EHD2-based molecules. Mean ± SD, *n* = 3.

	scFvCEA-EHD2	scFvCEA-EHD2-GT7	scFvCEA-EHD2-FT124	scFvCEA-EHD2-FT248
**calc. MW [kDa]**	83.0	90.0	100.6	121.6
**app. MW [kDa]**	109.5	244.4	255.4	420.5
**agg. temp. [°C]**	46	47	48	47
**ELISA [nM]**	0.3 ± 0.01	0.2 ± 0.01	0.2 ± 0.03	0.2 ± 0.03
**t_½_α [h]**	0.8 ± 0.18	0.4 ± 0.02	0.4 ± 0.03	2.4 ± 0.06
**t_½_β [h]**	5.0 ± 0.4	13.4 ± 1.4	17.0 ± 4.0	24.9 ± 2.7
**AUC [%×h]**	337 ± 45	290 ± 80	453 ± 54	1625 ± 159

**Table 4 molecules-27-03272-t004:** Biochemical, binding, and pharmacokinetic properties of scFv-EHD2-Cy5-based molecules. Mean ± SD, n.d.: not determined, *n* = 3.

	scFvFAP-Cy5	scFvFAP-EHD2-Cy5	scFvFAP-EHD2-FT124-Cy5	scFvFAP-EHD2-FT248-Cy5
**calc. MW [kDa]**	28.9	84.4	101.8	122.8
**app. MW [kDa]**	32.9	111.5	265.0	431.0
**FACS [nM]**	3.3 ± 0.03	0.4 ± 0.02	1.1 ± 0.05	1.1 ± 0.01
**t_½_α [h]**	0.5 ± 0.01	1.5 ± 1.0	0.4 ± 0.11	n.d.
**t_½_β [h]**	1.2 ± 0.03	2.6 ± 0.3	36.6 ± 9.4	39.3 ± 6.5
**AUC [%×h]**	98 ± 14	363 ± 69	1613 ± 234	2016 ± 142

## Data Availability

All processed data for the study are included within the manuscript and Appendix A. Raw datasets are available from the corresponding author on reasonable request.

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
