# Peer review of "GlycoTAIL and FlexiTAIL as Half-Life Extension Modules for Recombinant Antibody Fragments"

_molecules, 2022, doi:10.3390/molecules27103272_

Round 1
Reviewer 1 Report
The authors describe a new and original approach to extend the half-life of recombinant antibody fragments such as scFv Diabody or scFv-EHD2 modules by adding C-terminal amino acid cassettes named GlycoTAIL or flexiTAIL.
They describe in their paper convincing experiments demonstrating the full functionality of their different constructs by an ELISA or FACS approach.
They also demonstrate a dramatic increase in the half-life in blood of the scFv- EHD2 constructs after systemic injection in mice.
The article is very clear and well written.
Very very minor points
The authors should comment on the apparent affinity difference observed by FACS between scFv and scFv-EHD2 reflecting the avidity of the latter construct.
For clarity, the authors could add the theoretical molecular weight next to the constructs in Figures 2,3 and 4.
Author Response
Dear Reviewers,
thank you very much for reviewing our manuscript “GlycoTAIL and FlexiTAIL as half-life extension modules for recombinant antibody fragments” that has been submitted to the journal “Molecules”. We responded to each question and highlighted our response in green. The modifications within the manuscript are highlighted in blue.
The authors describe a new and original approach to extend the half-life of recombinant antibody fragments such as scFv Diabody or scFv-EHD2 modules by adding C-terminal amino acid cassettes named GlycoTAIL or flexiTAIL.
They describe in their paper convincing experiments demonstrating the full functionality of their different constructs by an ELISA or FACS approach.
They also demonstrate a dramatic increase in the half-life in blood of the scFv- EHD2 constructs after systemic injection in mice.
The article is very clear and well written.
We thank the reviewer for this kind comment.
Very very minor points
- The authors should comment on the apparent affinity difference observed by FACS between scFv and scFv-EHD2 reflecting the avidity of the latter construct.
Based on the FAP-targeting molecules, we used the monovalent (scFv) and bivalent (scFv-EHD2) binding molecules for targeting the tumor-stroma antigen FAP. The EC50 values of the scFv was strongly reduced (3.3 nM) compared to the bivalent scFv-EHD2 molecule (0.4 nM). Thus, we included the following sentence in the text:
Line 277:
The difference of EC50 of the monovalent (scFv) in comparison to the bivalent (scFv-EHD2, scFv-EHD2-FT128, scFv-EHD2-GT248) binding is significant and most likely due to avidity effect of the bivalent molecules. As the modified scFvFAP-EHD2-FT-Cy5 molecules also showed significant weaker binding than the unmodified scFvFAP-EHD2-Cy5 molecule, we believe this might be caused by sterical effects on antigen binding due to the chemical modification.
- For clarity, the authors could add the theoretical molecular weight next to the constructs in Figures 2,3 and 4.
Thank you very much for this suggestion. We have included the theoretical molecular weight in the Figures 2, 3, and 4.
Reviewer 2 Report
The paper entitled "GlycoTAIL and FlexiTAIL as half-life extension modules for recombinant antibody fragments" describes the potential for extending half-life of antibody fragments of different sizes, using several PEG mimetics.
The possibility of using tails of different length and different glycosylation potential makes this approach a more flexible solution than other strategies used, such as conjugation to PEG.
Several remarks should be addressed:
- The presence or absence of N-glycosylation in the different formats is highlighted in the text many times. Please explain what are the advantages or disadvantages of including glycosylation motifs in these molecules.
- The purpose of the modifications discussed here is to prolong half-life through increasing the size of the molecule. But at least for the larger formats, the size of the resulting molecule comes close to the size of a full IgG or a scFv-Fc. Please give some explanation why it is preferred over an Fc-containing antibody format.
- In Figure 6, please explain what are the individual dots. If these are the repetitions of each experiment, they do not fit the number indicated (n=3 for each experiment).
- There is no explanation how AUC was calculated. This should be added to the methods section and also its meaning should be explained more clearly in the text.
- Line 168: "all Db-based molecules showed similar binding to the cells…". The DbCEA format binding seems different than the others (Fig. 3E).
- Lines 171-172: "The analysis of the pharmacokinetic properties showed a clearly prolonged half-life for the GT7 and DbCEA-FT124 and DbCEA-FT248 compared to the DbCEA molecule". According to Figure 3 legend, only DbCEA-FT248 showed a significantly prolonged half-life (lines 163-164). The t½ values shown in table 2 also do not support this claim regarding GT7 and DbCEA-FT124. Please rephrase.
- Lines 178-179: " the EHD2 (IgE heavy chain domain 2) moiety serves as a covalently-linked dimerization module (Fig. 4A)". The actual structure of the molecule is nicely illustrated in Figure 1. It will be better to reference it here as well.
- Line 186: "… scFvCEA-EHD2-FT100…" should be FT124.
Author Response
Dear Reviewers,
thank you very much for reviewing our manuscript “GlycoTAIL and FlexiTAIL as half-life extension modules for recombinant antibody fragments” that has been submitted to the journal “Molecules”. We responded to each question and highlighted our response in green. The modifications within the manuscript are highlighted in blue.
The paper entitled "GlycoTAIL and FlexiTAIL as half-life extension modules for recombinant antibody fragments" describes the potential for extending half-life of antibody fragments of different sizes, using several PEG mimetics.
The possibility of using tails of different length and different glycosylation potential makes this approach a more flexible solution than other strategies used, such as conjugation to PEG.
Several remarks should be addressed:
1. The presence or absence of N-glycosylation in the different formats is highlighted in the text many times. Please explain what are the advantages or disadvantages of including glycosylation motifs in these molecules.
N-glycosylation was included in this study as a means of half-life extension, as shown previously for a single-chain Diabody comprising a GlycoTAIL (Stork et al., 2007). Due to its hydrophilic nature, N-glycans are highly soluble and able to increase the hydrodynamic radius, e.g. as also utilized in darbepoetin. For a comparison, the GlycoTAIL is used in this study further including the FlexiTAIL. The advantage is the use of this post-translational modification, i.e. it does not require chemical coupling, e.g. as used in PEGylated proteins. On the other sides, N-glycans are normally heterogeneous in composition, making it difficult to produce homogenous preparations.
2. The purpose of the modifications discussed here is to prolong half-life through increasing the size of the molecule. But at least for the larger formats, the size of the resulting molecule comes close to the size of a full IgG or a scFv-Fc. Please give some explanation why it is preferred over an Fc-containing antibody format.
The conjugation of small antibody fragments, like scFv or Db molecules, with moieties that increases the hydrodynamic radius results in proteins with prolonged circulation within the body. Modification of recombinant proteins with different GlycoTAIL or FlexiTAIL moieties allow to adjust the circulation time and the drug exposure of this molecule within the body. There are also possibilities in the field of Fc region modifications, however, the precise adjustment of the PK profile is not possible with Fc-containing molecules. Additionally, the Fc-part also contains further effector functions, like the activation of the immune system by binding to Fc-specific receptors. Our GlycoTAIL and FlexiTAIL moieties do not have any effector functions.
3. In Figure 6, please explain what are the individual dots. If these are the repetitions of each experiment, they do not fit the number indicated (n=3 for each experiment).
Figure 6 is a summary of all AUC and terminal half-life obtained in the different PK studies for the different molecules. Here, each symbol represents the mean of the terminal half-lives or the mean of the AUC of the different molecules. In the legend of Figure 6, we have included that this is the mean of the terminal half-lives or of the AUCs.
Line 326:
The apparent molecular weight was either analyzed against the mean of terminal half-life [h] or against the mean of the drug exposure (AUC[%*h]) of the different molecules.
4. There is no explanation how AUC was calculated. This should be added to the methods section and also its meaning should be explained more clearly in the text.
We have included the calculation of the AUC using an Excel add-in. Additionally, we explained that this describes the drug exposure of an applied molecule within the body.
Line 456:
PK analysis (t½α, t½β, AUC) was performed using Excel add-ins. The AUC describes the area under the curve and can be used as drug exposure of the applied molecule within the body.
5. Line 168: "all Db-based molecules showed similar binding to the cells…". The DbCEA format binding seems different than the others (Fig. 3E).
Indeed, the binding of the Db molecule to the LS174T cells was performed later compared to the other modified molecules. Due to changing and modification of the lasers within the flow cytometer machine, the intensity of the Db molecule was not the same as for the modified molecules. Here, we changed the y-axis of this graph from the relative MFI to the complete binding of each molecule.
6. Lines 171-172: "The analysis of the pharmacokinetic properties showed a clearly prolonged half-life for the GT7 and DbCEA-FT124 and DbCEA-FT248 compared to the DbCEA molecule". According to Figure 3 legend, only DbCEA-FT248 showed a significantly prolonged half-life (lines 163-164). The t½ values shown in table 2 also do not support this claim regarding GT7 and DbCEA-FT124. Please rephrase.
Thanks for this information. Indeed, the reviewer is right that there is a difference between the DbCEA-FT248 and the DbCEA molecules. Thus, we excluded DbCEA-GT7 and DbCEA-FT128 from this sentence.
Line 180:
The analysis of the pharmacokinetic properties showed a clearly statistical prolonged half-life for the DbCEA-FT248 compared to the DbCEA, DbCEA-GT7, and DbFT128 molecules (****).
7. Lines 178-179: " the EHD2 (IgE heavy chain domain 2) moiety serves as a covalently-linked dimerization module (Fig. 4A)". The actual structure of the molecule is nicely illustrated in Figure 1. It will be better to reference it here as well.
We have included the reference 23 (Seifert, O.; Plappert, A.; Fellermeier, S.; Siegemund, M.; Pfizenmaier, K.; Kontermann, R.E. Tetravalent antibody-scTRAIL fusion proteins with improved properties. Mol. Cancer Ther. 2014, 13, 101–111, doi:10.1158/1535-7163.MCT-13-0396.) to the results text as well as to the figure legend of Figure 1.
Line 75:
Antibody fragments are directed against carcinoembryonic antigen (CEA) and included a single-chain Fv fragment (scFv), a bivalent diabody (Db), and a bivalent scFv-EHD2 [23] fusion protein with calculated molecular masses of 26.9 kDa, 56.7 kDa, and 83.0 kDa, respectively (Fig. 1, Tables 1-3).
Line 88:
The scFv and diabody is show in light gray, EHD2 [23] is shown in gray.
8. Line 186: "… scFvCEA-EHD2-FT100…" should be FT124.
Thank you very much for indicating this mistyping. We have changed to scFvCEA-EHD2-FT124.
Line 194:
Single bands at ~70 kDa or ~100 kDa were detected for scFvCEA-EHD2-FT124 or scFvCEA-EHD2-FT248 molecule, respectively, under reducing conditions (Fig. 4B).
Reviewer 3 Report
The manuscript by Seifert and Kontermann describes the characterization of two modes that enables the extension of antibody fragments in the circulation of mice. The authors provides both biochemical and pharmacokinetics data that support their conclusions that these modifications retain the binding activities of the antibodies fragments and significantly extend their half-life in the circulation.
Below please find my comments and suggestion which I believe that should be incorporated in the revised version of the manuscript:
Line 124 – the affinity of 0.8nM seems to statistically differ from the other antibodies (0.5nM). Please refer to this point in the text and provide an explanation to this observation.
Line 132 – the statistical parameters are not provided (not only here but throughout the text).
Line 167 – these are major differences in the EC50 values. Please elaborate in the text.
Line 169 – the Bmax value of DbCEA is very low (50 MFI) when compared to the other antibodies (150 MFI). This seems problematic, as one would expect that the non-modified antibody will have at least the same binding characterization (not EC50) and Bmax.
Discussion: it is not clear what is the advantage (if any) of these modification over other modification of the same antibodies. Especially in terms of circulation half-life.
Line 340 – the composition of the N-glycans was not described in the results section.
Binding analysis (several figures, for example 2D, 2E): The authors should normalize the binding to the Bmax value of each antibody and present the data as percent of max binding. In this manner, it will be easier to compare the binding affinity.
PK studies: the curves should be fitted using non-linear regression (for example using two-phase decay or any other relevant method to fit the bi-phase clearance profile) and from that determine the Cmax value. Then, the curves should be plotted at percent of Cmax value. I guess it will also affect the T1/2 values throughout the text. This is the more accurate way to present and characterize PK data.
Figure 5D – Again, high variability in Bmax values. Normally, such differences (even though the EC50 is the same for the different antibodies) raise question about the availability of the detection antibody to recognize the bound molecule. Please repeat the binding experiments and/or provide detailed explanation to this observation.
Author Response
Dear Reviewers,
thank you very much for reviewing our manuscript “GlycoTAIL and FlexiTAIL as half-life extension modules for recombinant antibody fragments” that has been submitted to the journal “Molecules”. We responded to each question and highlighted our response in green. The modifications within the manuscript are highlighted in blue.
The manuscript by Seifert and Kontermann describes the characterization of two modes that enables the extension of antibody fragments in the circulation of mice. The authors provides both biochemical and pharmacokinetics data that support their conclusions that these modifications retain the binding activities of the antibodies fragments and significantly extend their half-life in the circulation.
Below please find my comments and suggestion which I believe that should be incorporated in the revised version of the manuscript:
- Line 124 – the affinity of 0.8nM seems to statistically differ from the other antibodies (0.5nM). Please refer to this point in the text and provide an explanation to this observation.
Thanks for this information about the statistics. We have included here the significant differences of the binding of the different molecules. We wrote that there is a significant reduced binding for the scFvCEA-FT248 molecule compared to the other analyzed molecules. We believe that this results from sterical hinderance of the FlexiTAIL128 moiety.
Line 127:
The scFvCEA-FT248 molecule showed significant reduced binding to the antigen compared to the other analyzed molecules (scFvCEA(*), scFvCEA-GT7(*), scFv-FT128(**)), most likely due to sterical hinderance of binding the antigen.
- Line 132 – the statistical parameters are not provided (not only here but throughout the text).
Thanks for the information about the reduced presentation of statistical parameters. We have included significant differences in all ELISA and FACS analysis, but also in the performed PK studies. We have included the significant differences between the analyzed molecules within the text.
Line 127:
The scFvCEA-FT248 molecule showed significant reduced binding to the antigen compared to the other analyzed molecules (scFvCEA(*), scFvCEA-GT7(*), scFv-FT128(**)), most likely due to sterical hinderance of binding the antigen.
Line 139:
Thus, scFvCEA-GT7 exhibited a terminal half-life of 1.4 hours, while the addition of the FlexiTAILs prolonged the terminal half-life to 5.7 (****) and 11.5 (****) hours for scFvCEA-FT124 and scFvCEA-FT248, respectively, which correlated with an increased drug exposure (AUC) of the half-life extended molecules, up to 2-, 5-, and 16-fold for scFvCEA-GT7 (***), scFvCEA-FT128 (****), and scFvCEA-FT248 (****), respectively (Table 1).
Line 174:
Significant differences were detected for the DbCEA compared to all other analyzed molecules (DbCEA-GT7 (***), DbCEA-FT128 (***), and DbCEA-FT248 (****)).
Line 180:
The analysis of the pharmacokinetic properties showed a clearly statistical prolonged half-life for the DbCEA-FT248 compared to the DbCEA, DbCEA-GT7, and DbFT128 molecules (****).
Line 207:
Statistical significance was detected for the scFvCEA-EHD2 molecule compared to the other analyzed molecules (scFvCEA-EHD2-GT7(***), scFvCEA-EHD2-FT148 (****), and scFvCEA-EHD2-FT248 (**)).
Line 229:
Significant differences were observed for the terminal half-life between the scFvCEA-EHD2 molecule and the other modified molecules (scFvCEA-EHD2-GT7 (*), scFvCEA-EHD2-FT128 (**), and scFvCEA-EHD2-FT248 (****)), while the scFvCEA-EHD2-FT248 was also significant to the scFvCEA-EHD2-GT7 (**) and the scFvCEA-EHD2-FT128 (*). For the AUC analysis, only the scFvCEA-EHD2-FT248 showed significant differences to the other analyzed molecules (****).
Line 286:
Significant differences were observed for both scFvFAP-EHD2-FT-Cy5 molecules compared to the unmodified molecules (****).
Line 289:
Here, the scFvFAP-EHD2-FT248-Cy5 showed a significant difference compared the other analyzed molecules (****).
- Line 167 – these are major differences in the EC50 values. Please elaborate in the text.
As described in the questions above, we have now included the significant differences also in this ELISA experiment. We have stated that there is a significant difference between the DbCEA molecules and the other modified DbCEA molecules.
Line 174:
Significant differences were detected for the DbCEA compared to all other analyzed molecules (DbCEA-GT7 (***), DbCEA-FT128 (***), and DbCEA-FT248 (****)).
- Line 169 – the Bmax value of DbCEA is very low (50 MFI) when compared to the other antibodies (150 MFI). This seems problematic, as one would expect that the non-modified antibody will have at least the same binding characterization (not EC50) and Bmax.
Thanks for this information about the FACS analysis of the Db molecules. Actually, there should be no difference between the unmodified Db and the modified DbCEA molecules. As the measurements were not performed on the same day, the exchange and modification of the laser of the flow cytometry machine did change the intensity of the different analyzed molecules. Thus, we have now normalized the values of our binding experiments for easier comparison.
- Discussion: it is not clear what is the advantage (if any) of these modification over other modification of the same antibodies. Especially in terms of circulation half-life.
Increasing the hydrodynamic radius of recombinant proteins resulted in a prolonged PK profile of the analyzed molecules. By choosing the correct length of GlycoTAIL or FlexiTAIL to the different molecules of interest, the circulation within the body but also the drug exposure of the molecules can be adjusted very precisely. By using antibodies, the Fc-part is responsible for the prolonged circulation of these molecules. Changing and modifying certain sequences within the Fc-part are critical, as this Fc-part also contains further effector functions like the activation of the immune system by binding to Fc-specific receptors.
- Line 340 – the composition of the N-glycans was not described in the results section.
We have included the composition of the GlycoTAIL in the beginning of the results in Line 71. Here, we describe the composition of this module.
Line 71:
For the GlycoTAIL module, the N-glycosylation sites are located 6 to 16 amino acids from each other.
- Binding analysis (several figures, for example 2D, 2E): The authors should normalize the binding to the Bmax value of each antibody and present the data as percent of max binding. In this manner, it will be easier to compare the binding affinity.
Thanks for this hint of the graphs. We have included the normalized data within each graph. Thus, we have modified the Figure 2, 3, 4, and 5 and included the normalized binding of the different molecules. Additionally, we have also analyzed the EC50 values of the different experiments and updated the values.
- PK studies: the curves should be fitted using non-linear regression (for example using two-phase decay or any other relevant method to fit the bi-phase clearance profile) and from that determine the Cmax value. Then, the curves should be plotted at percent of Cmax value. I guess it will also affect the T1/2 values throughout the text. This is the more accurate way to present and characterize PK data.
The PK parameters were calculated by an add-in of Excel, which computes the initial half- life (t½α), the terminal half-life (t½β), and the drug exposure (AUC) of the different molecules. This tool does not allow to perform a non-linear regression. This add-in was used in previous studies for PK analysis. Another tool (PKSolver) was also not able to fit data using non-linear regression.
- Figure 5D – Again, high variability in Bmax values. Normally, such differences (even though the EC50 is the same for the different antibodies) raise question about the availability of the detection antibody to recognize the bound molecule. Please repeat the binding experiments and/or provide detailed explanation to this observation.
We have also included the normalized graph of the flow cytometry analysis of the different FAP-targeting molecules. Additionally, we have also included the significant difference of the binding of the different analyzed molecules.
Line 277:
The difference of EC50 of the monovalent (scFv) in comparison to the bivalent (scFv-EHD2, scFv-EHD2-FT128, scFv-EHD2-FT248) binding is significant (****) and most likely based on the avidity effect of the bivalent molecules. As the modified scFvFAP-EHD2-FT-Cy5 molecules also showed significant worse binding (****) than the unmodified scFvFAP-EHD2-Cy5 molecule, we believe this might be sterical hinderance of antigen binding.